# Nasal Cytology on 241 Children: From Birth to the First 3 Years of Life and Association with Common Airways Diseases

**DOI:** 10.3390/jpm13040687

**Published:** 2023-04-19

**Authors:** Cecilia Rosso, Federica Turati, Alberto Maria Saibene, Elvira Verduci, Emanuela Fuccillo, Maria Chiara Tavilla, Mauro Magnani, Giuseppe Banderali, Monica Ferraroni, Eugenio De Corso, Giovanni Felisati, Carlotta Pipolo

**Affiliations:** 1Department of Otorhinolaryngology, Santi Paolo e Carlo Hospital, Università degli Studi di Milano, 20148 Milan, Italy; 2Department of Health Sciences, Università degli Studi di Milano, 20148 Milan, Italy; 3Branch of Medical Statistics, Department of Clinical Sciences and Community Health, Università degli Studi di Milano, 20122 Milan, Italy; federica.turati@unimi.it (F.T.);; 4Pediatric Department, “Vittore Buzzi” Children’s Hospital, 20154 Milan, Italy; 5Department of Hand Surgery and Microsurgery, Gaetano Pini—CTO Orthopaedic Institute, University of Milan, 20122 Milan, Italy; mauro.magnani@unimi.it; 6Pediatric Department, Santi Paolo e Carlo Hospital, Università degli Studi di Milano, 20148 Milan, Italy; 7Fondazione IRCCS Ospedale Maggiore Policlinico, 20122 Milan, Italy; 8Department of Otorhinolaryngology, La Fondazione Policlinico Universitario Agostino Gemelli IRCCS, Universita Cattolica del Sacro Cuore, 00168 Rome, Italy

**Keywords:** nasal cytology, children, newborns, otitis, asthma, URTI, rhinosinusitis, rhinitis, pediatric

## Abstract

Background: Nasal cytology at birth and in the pediatric age is barely investigated regarding its association with the onset of common pediatric diseases. Methods: We enrolled 241 newborns within their first 24 h of life, studying their nasal cellular composition and repeating this at 1 and 3 years of life. We collected anamneses of perinatal factors and external factors (parental smoking, passive smoking, breastfeeding), and the prevalence of otitis, rhinosinusitis, bronchitis, asthma, and allergy at all timepoints. Results: 204 children completed the study. At birth, there was a prevalence of ciliated cells and rare neutrophils. At 1 and 3 years, ciliated cells started reducing in favor of muciparous cells and neutrophils. We found that caesarian delivery and nasogastric tube usage for choanal patency are significantly related to a certain cellular nasal composition. Additionally, development of upper respiratory tract infections, AOM (acute otitis media) and allergy correlates with specific cytological compositions which may predict those pathologies. Conclusions: Our study is the first to show the normal nasal mucosa cellular composition and development in the first 3 years of life in a large cohort. Nasal cytology may be a tool for early risk assessment in the occurrence of upper airway disease.

## 1. Introduction

Pediatric diseases such as acute otitis media (AOM), upper respiratory tract infections (URTI), bronchitis, asthma, allergy, rhinosinusitis, and adenoidal hypertrophy are a daily challenge for clinicians such as otorhinolaryngologists, pediatricians, and allergists.

Those pathologies may lead, especially during the growth period, to relevant comorbidities and complications such as obstructive sleep apnea (OSA) and hearing impairment, as well as learning and attention difficulties [1] and it is presumable that such diseases may have important implications in both scholastic and social performances [2].

The possibility to screen newborns for the risk of developing these diseases would be therefore of enormous value in order to potentially engage in strict follow-up that could guarantee early diagnosis and hence satisfactory treatment. However, to our knowledge, it is currently difficult to determine predictive factors which may increase the risk of developing any of these clinical issues. Nowadays family history, and smoking by the parents and similar environmental factors, are the only correlations that clinicians may find which may suggest a predisposition to some of these pathologies such as asthma or allergy. However, no objective measures have been introduced into neonatal screenings.

Nasal cytology (NC) is a simple diagnostic procedure that evaluates the health of the nasal mucosa by recognizing and counting cell types and their morphology [3,4]. NC may detect infectious agents such as fungi and bacteria, allowing for example the diagnosis of infectious rhinitis. It also evaluates cellular composition, detecting ciliated and muciparous cells, eosinophils, neutrophils, mast cells, and others. Specific cytological patterns in NC can help in discriminating among various forms of rhinitis, including allergic rhinitis (AR), non-allergic rhinitis (NAR), idiopathic rhinitis, and overlapping forms [4].

Since the diagnosis of atopy and allergy in children, especially in the youngest age groups, is a difficult challenge and requires careful and broad analysis, NC may be considered as a valid diagnostic tool, thanks to its simple, noninvasive, and inexpensive method which is able to show any sign of local type-2 inflammation in the nasal mucosa [5]. Moreover, following the “united airways” concept, both upper and lower airway tracts share the same mucosal structure and functioning, and nasal secretions are delivered directly into the bronchial airways [6]. Therefore, diagnosis and treatment of nasal airways is essential to also improve and/or prevent respiratory tract disfunctions [6].

As early as in 1985 Cohen et al. used nasal cytology to study the nasal mucosa of 22 infants, concluding that the technique is as safe and effective in infants as in children and adults and that the majority of healthy young infants do not have nasal eosinophilia [7].

The use of nasal cytology in children is becoming increasingly present, and a recent study has even evaluated the accuracy of the cotton nasal swab (NSW) as a sampling method in children to identify nasal cytotypes and rhinitis phenotypes, using nasal scraping (NSC) for comparison: even though NSW has been shown to be slightly more tolerable in a pediatric population, it has a significantly inferior diagnostic accuracy compared to nasal scraping [8].

A study performed by Tarchalska-Kryńska et al. evaluated nasal mucosa with cytology in healthy newborns: cytograms of the nasal mucosa differ from these of adults and older children, with a prevalence of columnar cells or neutrophils [9]. An interesting study by Kròl and colleagues evaluated nasal cytology of newborns who were exposed to tobacco smoke during fetal life: the most common type of cytogram contained neutrophils, columnar cells, and squamous cells. Therefore, analysis revealed that active and passive smoking during pregnancy does not influence the cytological picture of the nasal mucosa of neonates [10].

We thought to extend this concept to other common pediatric pathologies, and hence to uncover if there is a link between nasal cellular composition at birth and the chance of developing such diseases in the first 3 years of life. The literature has already explored the ability of nasal cytology for predicting the onset of various diseases [11], but to our knowledge no longitudinal prospective studies have been performed correlating nasal cytology at birth and the later onset of any disease.

This study aims to collect and analyze the nasal mucosa cytological composition, on a bigger scale as previously reported, at birth (in the first 24 h of life) and therefore obtain an uncontaminated and representative sample, followed by collection of nasal mucosa cytology at 1 and 3 years of life in order to analyze any variation on its composition. Moreover, we set a longitudinal prospective study until 3 years of life of the child in order to evaluate the association of nasal cytology with the development of diseases such as asthma, rhinitis, allergy, bronchitis, otitis, and URTI, and to assess whether nasal cytology is influenced by selected external factors. Finally, we would like to provide a baseline of cytological composition as well as any external influencing factors that could interfere with the nasal cell composition in order to further investigate the relationship between genetic and environmental components in the pathogenesis of these diseases.

## 2. Materials and Methods

A longitudinal prospective study was carried out. We enrolled 241 consecutive newborns whose mothers were admitted for delivery at San Paolo Hospital, Milan from February to December 2016. Patients underwent nasal cytology at birth, and at 1 year and at 3 years of life.

### 2.1. Inclusion and Exclusion Criteria and Data Collection

We included newborns of both sexes born after the 34th pregnancy week, with both parents Italian, with the rationale to minimize genetic differences due to diverse ethnic backgrounds. Newborns with any pathology and/or alterations of the head and neck area were excluded.

All participating newborns received nasal scraping within their first 24 h. According to standardized criteria, samples have been collected by means of a Rhino-Probe™ curette (Arlington Scientific Inc. Springville, Utah, USA) from the middle portion of the inferior turbinate where the rate ciliate/mucinous cells are expected to be well balanced [12]. The procedure was performed under anterior rhinoscopy with an appropriate light source. No application of anesthetic is required. The presence of squamous cells usually indicates a contamination from the skin epithelium of the nares, thus a not-optimal sampling. The curette was immediately smeared on a glass slide, with attention paid to properly distributing the collected material on the slide and dissipating the possible clots of mucus. Then, samples were stained with May-Grunwald-Giemsa (MGG) Quick Stain coloring (Bio -Optica) [3]. The slides were examined by a Zeiss Axio Lab A1, and 50 microscopic fields were read at a magnification of 1000x to assess the presence of normal and abnormal cellular elements. A minimum of fifty fields is considered necessary to identify a sufficient number of cells. A qualitative and quantitative grading was used [8,13].

The process was repeated at 1 and 3 years of life. A skin prick test for frequent allergens was performed during the last sampling (3 y) in order to discern AR from NAR and to objectify allergic sensitization.

Structured questionnaires were administered face-to-face to newborns’ parents at birth, 1 year, and 3 years of life. At birth, we collected socio-economic factors, anamnestic data of parents, pregnancy exposures and delivery characteristics (e.g., pregnancy complication, drug and supplement use, mother’s alcohol use, type of delivery, induction and delivery complications), parent smoking (including mother’s passive smoking during pregnancy), and characteristics of the children at birth. At 1 and 3 years of age we collected data on siblings, breastfeeding and solid foods, children’s exposure to passive smoking, and kindergarten attendance. In addition, children’s history of AOM, URTI, bronchitis, and bronchial asthma/wheezing during, respectively, the first and the second/third years of life were reported by their parents.

Since there is no validated questionnaire evaluating the incidence of asthma, URTI, allergy, bronchitis, and AOM all together, colleagues built a questionnaire including questions taken from the ISAAC study for asthma and allergies, and The Acute Respiratory Tract Infection Questionnaire for URTI, AOM, and bronchitis [13,14]. Questionnaires were administered in the Italian language, since an inclusion criteria for patients’ selection was having Italian parents. Moreover, parents were asked to bring to the 1- and 3-year evaluation all documentation relating to primary care visits, hospital stays, and specialists’ visits.

At year 3, NC was performed only on 118 patients due to SARS-COV2 pandemic limitations at our hospital; 207 underwent questionnaire completion (parents and their children who could not come to the hospital because of the aforementioned limitations underwent telephonic questionnaire completion).

### 2.2. Statistical Analysis

Descriptive statistics were used to describe data on newborns, parents, and nasal mucosa cytological composition.

Sex-adjusted log-binomial regression models were used to evaluate the association of selected exposures (in pregnancy, at birth, and during the first 3 years of the child’s life) with nasal mucosa cytological composition at birth, 1 year, and 3 years. Relative risks (RR) with their corresponding 95% confidence intervals (CI) were calculated. In case of cross-sectional associations, we calculated the odds ratios (OR) and the corresponding 95% CI by sex-adjusted logistic regression models.

Log-binomial regression models adjusted for sex were also used to assess the association between nasal mucosa cytological composition at birth (exposure) and the development of AOM, URTI, bronchitis, and bronchial asthma/wheezing (outcomes) during the first year of life and during the first 3 years of life. Similarly, we estimated the RR of positivity to the skin prick test at the 3-year evaluation according to NS at birth.

In Appendix A analyses, we evaluated the association between the presence of selected diseases during the first year of life and nasal mucosa cytological composition at 1 year using the chi-square test or the Fisher exact test.

All the analyses were performed using the SAS software, version 9.4 (SAS Institute, Inc., Cary, NC, USA).

## 3. Results

We included 241 newborns in our study; among them, 233 had sufficient material to evaluate NC at birth; at 1 year, questionnaire data were available for 222 children. Among them, 218 had sufficient material to evaluate NC. At 3 years, 204 children had questionnaire data and 126 had NC data. Overall, a group of 118 children completed the follow-up period for up to 3 years, fulfilling all the evaluated fields, including NC, while 86 patients were evaluated for up to 3 years without performing NC due to pandemic restrictions, with a total case series of 204 children who completed the study (Appendix A).

A description of newborns and their parents’ characteristics as well as pregnancy and delivery characteristics is shown in Table 1. Overall, 50.6% of newborns were males; the mean gestational age at birth was 39.2 weeks (SD 1.3). Standard delivery was accomplished by 77.1% of mothers, while 22.9% of children were born through cesarian delivery.

### 3.1. Results

#### 3.1.1. Nasal Mucosal Composition

Cellular composition of nasal mucosa at birth, at 1 year, and at 3 years is synthetized in Table 2. At birth, there was a prevalent cellular composition of ciliated cells and rare neutrophils. At 1 year, ciliated cells started reducing in favor of muciparous cells and neutrophils; rare bacteria were detectable. Evaluation at 3 years showed a stable composition with a progressive decrease per field in ciliated cells and an increase in bacteria.

#### 3.1.2. Factors Influencing Nasal Mucosal Composition

Table 3 gives results for the association between selected exposures and nasal mucosa composition at each time point. Caesarian delivery significantly impacted the presence of eosinophils at birth: 11.5% of children born by caesarean delivery vs. 2.2% of those born by vaginal delivery had eosinophils at birth, for a corresponding RR of 5.17 (95% CI: 1.52–17.6). No association was found between delivery modus and presence of bacteria at rhinocytogram.

Evaluation of choanal patency at birth with nasogastric tube was associated with the presence of eosinophils (8.6% vs. 2.0%, RR: 4.66, 95%, CI: 1.23–17.64) and abundant neutrophils (61.7% vs. 43.6%, RR: 1.37, 95% CI: 1.07–1.77) at birth; the use of nasogastric tube was also associated with a lower risk of abundant muciparous cells at birth (25.9% vs. 36.7%, RR: 0.66, 95% CI: 0.44–1.01) (Table 3). No other factor was significantly associated with nasal mucosa composition at the various time points.

#### 3.1.3. Nasal Mucosal Composition and Disease Development

Results for the association between nasal cytology at birth and the development of AOM, URTI, bronchitis, and bronchial asthma/wheezing during the first year and during the first 3 years of life are summarized in Table 4.

Among the 214 children who had both nasal cytology data at birth and questionnaire data at 1 year, 32 were reported to have had AOM during the first year of life (15.0%), 196 URTI (91.6%), 58 bronchitis (27.1%), 14 allergy (6.5%), and 23 bronchial asthma/wheezing (10.7%). Presence of eosinophils at birth significantly increased the risk of AOM during the first year of life (RR = 3.4, 95% CI: 1.55–7.62). No other significant association was found between the various cellular types in the nasal mucosa at birth and diseases occurrence during the first year.

Within 3 years of life, AOM developed in 43.5% of children (87 out of 200 with available information on nasal cytology at birth and outcome), URTI in 94.4% (201 out of 213), bronchitis in 46.5% (93 out of 200), and bronchial asthma/wheezing in 19.9% (39 out of 196); 17 out of the 152 children who underwent skin-prick testing at the 3-year visit were positive for at least one allergen (18.2%).

Nasal cytology at birth did not significantly affect the risk of the various diseases during the first 3 years of life: in particular, the direct association between the presence of eosinophils at birth and the risk of AOM was no longer detectable (RR: 1.32, 95% CI: 0.72–2.44) (Table 4).

#### 3.1.4. Influence of Diseases on Nasal Mucosal Composition

Children with URTI or allergy during the first year of life had lymphocytes more frequently at nasal cytology evaluation at 1 year (URTI: 23.5% vs. 0%, *p* = 0.015; allergy: 42.9% vs. 20.1%, *p* = 0.084).

Children with allergy during the first year of life also tended to have abundant (++/+++/++++) neutrophils more frequently (100% vs. 67.7%, *p* = 0.084), and those with bronchial asthma/wheezing to have abundant muciparous cells less frequently (16% vs. 33.7%, *p* = 0.074) (Appendix A).

## 4. Discussion

The cytological expression of nasal mucosa in newborns and its association with the onset of ENT and other disorders is barely investigated.

Nowadays, we can consider the rhinocytogram in adults to be well studied: normal nasal mucosa is composed of ciliated and muciparous cells, with a ratio of 4/5:1, plus straited and basal cells. Other cellular types should not be present, except for rare neutrophils. The evidence of eosinophils, mast cells, bacteria, and macrophages has to be considered a sign of nasal pathology [3].

### 4.1. Rhinocytogram

Our study shows how, at birth, nasal cellular composition appears to have a preponderance of ciliated cells and almost an absence of muciparous cells, with some neutrophils. At 1 year of life cellular composition seems to shift towards the normal adult composition, with a slight decreasing of ciliated cells in favor of muciparous cells (ratio 4:1); neutrophils, however, are the only pathological cells which remain higher than in adult life. This could be explained by the higher incidence of infectious rhinitis and adenoiditis in children of 1–3 years and the consequent activation of neutrophils in the fight for bacterial growth. Evaluation at 3 years of life shows a stability of cytological assets.

A study performed by Tarchalska-Kryńska et al. evaluated nasal mucosa with cytology in healthy newborns and cytograms of the nasal mucosa of 1–7-day-old infants and showed how those differ from those of adults: both columnar cells and neutrophils are prevalent in cytograms of the healthy newborn children [9]. Despite the significance of the study, inclusion criteria allowed newborns of up to 7 days of life, whose nasal mucosa was already exposed to a multitude of external influxes of the environment. This could possibly alter the evidence of normality of an otherwise untouched mucosa.

In addition, premature newborns seem to respect the same cellular composition [15]: we did not evaluate this aspect since the original inclusion criteria allowed only to-term infants, nevertheless it would have been an interesting factor to consider.

### 4.2. External Factors

It is interesting to understand if there are any external factors during pregnancy and after birth related to the onset of altered nasal cytology at birth, which could be an alarm bell for the development of further diseases. We analyzed whether there is an association with external and parental factors (e.g., breastfeeding at birth and at 1 year, smoking).

An interesting study by Krol and colleagues evaluated nasal cytology of newborns who were exposed to tobacco smoke during fetal life: the most common type of cytogram contained neutrophils, columnar cells, and squamous cells. Analysis revealed that active and passive smoking during pregnancy do not influence the cytological picture of the nasal mucosa of neonates [10] as shown in Table 3: it remains comparable to the one of the whole case series.

### 4.3. Perinatal Influencing Factors

Analysis of pregnancy and delivery aspects revealed how caesarian delivery impacts the presence of eosinophils at birth, carrying the possibility of developing eosinophils-driven diseases. This may be also due to different handling of the child in the first minutes to hours after birth, as often those deliveries are performed for more high-risk situations and/or the absence of the passage of the birth canal. The literature reinforced our evidence: a study by Jensen et al. proved how caesarian delivery is associated with a higher risk of developing eosinophilic esophagitis [16]. Additionally, Ren and his group discovered how cesarian delivery aggravates the nasal symptoms of AR in mice [17]. Although we did not investigate esophagitis and AR, our analysis did not find any significant association between cesarian delivery and the incidence of atopic diseases such as asthma or allergies, or rhynocytograms that could be those of allergic rhinitis patients.

Further investigation about this association in a wider population and with a longer follow-up would be very important to validate or rebut our results.

Moreover, evaluation of choanal patency at birth with nasogastric tube increased the chance to have eosinophils and abundant neutrophils in nasal mucosa; to our knowledge, no studies have evaluated this aspect previously. We suppose that the stimulus given by the nasogastric tube at birth may lead to an inflammatory reaction with a recruitment of eosinophils and neutrophils in nasal mucosa even right after birth, showing the quick response extra-utero towards mechanic stimuli; nevertheless, it does not carry any further consequences in the evaluated time period. Interestingly, delivery modus (vaginal vs. cesarian) did not influence presence and quantity of bacteria at rhinocytogram at birth; however, microscopic analysis lacks the potential to differentiate distinctive types of bacteria, a fact that we would expect to differ between the two populations. Microbiome analysis could shed light on this.

### 4.4. Association of Rhinocytogram and ENT Diseases

Alteration of the rhinocytogram appears to be associated with the onset of a spectrum of ENT diseases.

### 4.5. Otitis

Our analysis revealed that the presence of eosinophils in nasal mucosa at birth is associated with a higher risk of developing AOM at 1 year; among these patients, only one dropped out of the study at the 3-year follow-up. All others corroborated how that risk disappears at 3 years. The literature does not provide any study about AOM incidence associated to a newborn’s nasal cytology; therefore, further studies would be interesting to validate our finding and hence to investigate how the growth process up to 3 years may improve this predisposition.

### 4.6. Atopy

When correlating nasal cytology to atopy, the literature seems divided. Borres analyzed cytograms of infants with a positive family history of allergies and then repeated the study at 18 months of age. In children with symptoms of atopy mast cells were more common. Eosinophils were in turn found in both groups of children without significant association [18]. Other authors maintain that eosinophils in nasal mucosa are predictors of the occurrence of varying atopic diseases (atopic dermatitis, asthma, and AR) [19,20,21].

Another interesting report about the development of mast cells and nasal eosinophils from age 4 months through 4 years in children of atopic parents reported that both eosinophils and mast cells were rare at 4 months in all infants, increased in atopic children from 1 to 4 years, and remained infrequent in nonatopic children [19]. In addition, Kajosaari et al. proved the higher percentage of mast cells and eosinophils in atopic children [22].

Surprisingly, we did not find any significant association of nasal cytology with the onset of atopy. In fact, none of the patients who were positive to skin prick tests at 3 years presented eosinophils at nasal cytology at birth or at 3 years of age, while only one child with a history of allergy presented eosinophils at birth.

On the other hand, this is the first study finding an association between allergy during the first year of life and the presence of lymphocytes in nasal cytology at 1 year. It would be interesting to further investigate this finding, expanding the observation of our study to a longer follow-up and to a wider cohort.

### 4.7. URTI

Also, the occurrence of URTI during the first year of life appeared to be related to the presence of lymphocytes in nasal cytology at 1 year. The literature does not provide any evidence about this association. Other associations of the rhinocytogram and URTI at different timepoints were not seen.

### 4.8. Asthma

When evaluating lower airways, we found an association between asthma and wheezing during the first year of life and the number of muciparous cells in nasal mucosa at 1 year. A higher number of muciparous cells at 1 year could be a consequence of the absence of asthma and wheezing during the first year of life or may protect against the disease; in any case, the association was not statistically significant. The literature shows how the nose might be envisaged as the open window of the lower respiratory tract [22]. Additionally, among children, there is evidence that nasal epithelial cells are a valid surrogate for bronchial epithelial cells, although studies have been more focused on other mediators such as interleukins, vascular endothelial growth factor (VEGF), interferon-b (IFN-b), and transforming growth factor-b (TGF-b), as well as matrix metalloprotein-9 (MMP-9), and not on nasal cellular composition [22].

Our study carried interesting findings: we could assess the normal cellular composition of newborns’ nasal mucosa, which is still poorly explored. On the other hand, it led us to depart from the diffuse thought that nasal eosinophils usually correlate with atopy and are already present at birth, since in our study neither nasal cytology at birth or at 3 years was associated with the onset of allergy, URTI, bronchitis, or asthma and only the presence of eosinophils at birth increased the risk of AOM during the first year of life.

Wider studies with longer follow-ups are needed to confirm our results. Our limited number of patients, also due to the unforeseeable advent of the SARS-COV2 pandemic, and the often low prevalence of specific cells at rhinocytogram, resulted in low statistical power to detect many of the associations.

In addition, the inclusion of premature newborns could have provided interesting findings, as well as the evaluation of nasal mucosa in association with the onset of AR, adenoidal hypertrophy, or obstructive sleep disorders.

## 5. Conclusions

Nasal cytology has been proved to be a promising tool for risk stratification. In our study, we found that caesarian delivery and nasogastric tube usage for testing choanal patency are significantly related to the development of a certain cellular nasal composition. Additionally, some specific nasal cytologies were shown to be statistically significant in the prediction of the onset of further airway diseases such as upper respiratory tract infections, AOM, and allergy.

However, in our study no clearcut associations could be shown: further studies are essential to validate its risk-evaluation role in the pediatric population.

## Figures and Tables

**Table 1 jpm-13-00687-t001:** Characteristics of 241 newborns included in the study and their parents.

	N ^a^ (%)
Male sex	122 (50.6)
Gestational age (week), mean (SD)	39.2 (1.3)
Birth weight (g), mean (SD)	3266 (413)
Choanal exploration with nasogastric tube	84 (35.1)
Siblings	118 (53.4)
Kindergarten ^b^	78 (35.1)
Exclusive breastfeeding ^c^	136 (61.3)
Maternal age at birth (yr), mean (SD)	33.7 (5.0)
Paternal age at birth (yr), mean (SD)	36.5 (5.7)
University maternal education	103 (42.9)
University paternal education	82 (34.3)
Mother passive smoking during pregnancy	39 (16.3)
Any pregnancy complication	78 (32.5)
Delivery	
Vaginal	186 (77.1)
Caesarean section	55 (22.9)
Elective	37 (67.9)
Emergency	18 (32.1)

^a^ Numbers in the Table are frequencies (percentages), unless otherwise specified. The sum may not add up to the total because of missing values. Percentages are calculated over the number of subjects with non-missing information. ^b^ During the first year of life. ^c^ Until solid food introduction.

**Table 2 jpm-13-00687-t002:** Nasal mucosa cytological composition at birth (N = 233), 1 year (N = 210 ^a^), and 3 years (118 ^b^).

	N ^c^ (%)		
	Birth	1 year ^a^	3 year ^b^
Ciliated cells			
0	7 (3.0)	2 (1.0)	5 (4.2)
+	15 (6.4)	35 (16.7)	21 (17.8)
++	25 (10.7)	55 (26.2)	29 (24.6)
+++	54 (23.2)	50 (23.8)	38 (32.2)
++++	132 (56.7)	68 (32.4)	25 (21.2)
Muciparous cells			
0	152 (65.2)	80 (38.1)	50 (42.4)
+	52 (22.3)	63 (30.0)	47 (39.8)
++	21 (9.0)	47 (22.4)	18 (15.3)
+++	6 (2.6)	14 (6.7)	2 (1.7)
++++	2 (0.9)	6 (2.9)	1 (0.9)
Neutrophils			
0	36 (15.5)	25 (11.9)	14 (11.9)
+	80 (34.5)	40 (19.1)	30 (25.4)
++	56 (24.1)	46 (21.9)	26 (22.0)
+++	27 (11.6)	52 (24.8)	28 (23.7)
++++	33 (14.2)	47 (22.4)	20 (17.0)
Eosinophils			
0	223 (95.7)	195 (93.3)	110 (93.2)
+	10 (4.3)	10 (4.8)	6 (5.1)
++	0	3 (1.4)	2 (1.7)
+++	0	1 (0.5)	0
Lymphocytes			
0	204 (88.3)	165 (78.6)	110 (85.6)
+	25 (10.8)	32 (15.2)	15 (12.7)
++	2 (0.9)	13 (6.2)	2 (1.7)
Mast-cells			
0	232 (99.6)	208 (99.0)	118 (100.0)
+	1 (0.4)	2 (1.0)	0
Macrophages			
0	231 (99.6)	191 (100.0)	118 (100.0)
+	1 (0.4)	0	0
Bacteria			
0	190 (81.9)	88 (41.9)	50 (42.4)
+	30 (12.9)	42 (20.0)	38 (32.2)
++	8 (3.5)	42 (20.0)	14 (11.9)
+++	3 (1.3)	26 (12.4)	11 (9.3)
++++	1 (0.4)	12 (5.7)	5 (4.2)

Semiquantitative evaluation of cellular composition is based on the Evaluation table of “Atlante di citologia nasale. M. Gelardi.” [3]. ^a^ The analysis included only children with nasal cytology data both at birth and at 1 year. ^b^ The analysis included only children with nasal cytology data at birth, at 1 year and at 3 years. ^c^ The sum may not add up to the total because of missing values. Percentages are calculated over the number of subjects with non-missing information.

**Table 3 jpm-13-00687-t003:** Association of type of delivery, nasogastric tube, exposure to smoking, and breastfeeding with nasal mucosa cytological composition at birth, 1, and 3 years of life.

	Muciparous Cells		Neutrophils		Eosinophils		Lymphocytes		Bacteria	
	Abundant ^e^ n (%)	RR ^f^ (95% CI)	Abundant ^e^ n (%)	RR ^f^ (95% CI)	Present n (%)	RR ^f^ (95% CI)	Present n (%)	RR ^f^ (95% CI)	Present n (%)	RR ^f^ (95% CI)
NC at Birth										
Maternal Smoking in Pregnancy ^a^										
**No**	66 (35.7)	1	92 (49.7)	1	10 (5.4)		23 (12.6)	1	34 (18.4)	1
**Yes**	14 (29.8)	0.83 (0.52–1.35)	24 (52.2)	1.05 (0.77–1.42)	0 (0)	ne	4 (8.5)	0.68 (0.25–1.87)	8 (17.4)	0.95 (0.47–1.90)
**Delivery**										
**Vaginal**	63 (35.0)	1	87 (48.6)	1	4 (2.2)		19 (10.6)	1	34 (19.0)	1
**Caesarean section**	17 (32.7)	0.94 (0.61–1.46)	29 (55.8)	1.15 (0.87–1.52)	6 (11.5)	5.17 (1.52–17.6)	8 (15.7)	1.48 (0.69–3.18)	8 (15.4)	0.81 (0.40–1.64)
**Choanal exploration with Nasogastric tube**										
**No**	58 (36.7)	1	65 (43.6)	1	3 (2.0)	1	14 (9.4)	1	30 (20.1)	1
**Yes**	21 (25.9)	0.66 (0.44–1.01)	50 (61.7)	1.37 (1.07–1.77)	7 (8.6)	4.66 (1.23–17.64)	12 (15.0)	1.65 (0.79–3.42)	10 (12.4)	0.61 (0.32–1.20)
**NC at 1 year**										
**Smoking exposure ^b^**										
**No**	48 (32.2)		104 (69.8)	1	12 (8.1)	1	34 (22.8)	1	92 (61.7)	1
**Yes**	21 (30.9)	0.96 (0.63–1.47)	47 (69.1)	0.98 (0.81–1.18)	2 (3.0)	0.37 (0.09–1.61)	13 (19.1)	0.84 (0.47–1.48)	34 (50.0)	0.81 (0.62–1.06)
**Delivery**										
**Vaginal**	52 (31.1)	1	119 (71.3)	1	11 (6.6)	1	38 (22.8)	1	95 (56.9)	1
**Caesarean section**	17 (34.0)	1.08 (0.69–1.69)	32 (64.0)	0.89 (0.71–1.11)	3 (6.0)	0.90 (0.26–3.12)	9 (18.0)	0.79 (0.41–1.51)	31 (62.0)	1.09 (0.85–1.41)
**Nasogastric tube**										
**No**	42 (30.2)		101 (72.7)	1	10 (7.3)	1	30 (21.6)	1	84 (60.4)	1
**Yes**	25 (32.5)	1.10 (0.73–1.65)	50 (64.9)	0.91 (0.75–1.08)	4 (5.2)	0.72 (0.23–2.21)	17 (22.1)	1.04 (0.61–1.76)	42 (54.6)	0.91 (0.71–1.16)
**Exclusive breastfeeding ^c^**										
**No**	28 (32.9)		57 (67.1)	1	4 (4.7)	1	19 (22.4)		52 (61.2)	1
**Yes**	41 (30.8)	0.95 (0.64–1.41)	95 (71.4)	1.05 (0.87–1.26)	10 (7.6)	1.61 (0.52–4.98)	28 (21.1)	0.94 (0.56–1.58)	75 (56.4)	0.92 (0.74–1.16)
**Breastfeeding at 1 year**										
**No**	50 (36.0)		98 (70.5)	1	10 (7.3)		33 (23.7)	1	80 (57.6)	1
**Yes**	19 (24.4)	0.57 ^g^ (0.31–1.07)	54 (69.2)	0.95 ^g^ (0.52–1.74)	4 (5.1)	0.69 ^g^ (0.21–2.29)	14 (18.0)	0.70 ^h^ (0.35–1.42)	47 (60.3)	1.12 ^h^ (0.64–1.70)
**NC at 3 years**										
**Smoking exposure ^d^**										
**No**	17 (20.2)	1	52 (61.9)	1	8 (9.5)	1	13 (15.5)	1	50 (59.5)	1
**Yes**	6 (14.6)	0.71 (0.30–1.65)	23 (56.1)	0.92 (0.67–1.27)	1 (2.4)	0.26 (0.03–1.99)	4 (9.8)	0.63 (0.22–1.81)	21 (51.2)	0.90 (0.64–1.26)
**Delivery**										
**Vaginal**	19 (18.6)	1	63 (61.8)	1	9 (8.8)		16 (15.7)	1	59 (57.8)	1
**Caesarean section**	5 (20.8)	1.03 (0.43–2.46)	13 (54.2)	0.91 (0.61–1.36)	0 (0)	ne	1 (4.2)	0.27 (0.04–1.91)	13 (54.2)	0.96 (0.65–1.42)
**Nasogastric tube**										
**No**	19 (23.5)	1	52 (64.2)	1	8 (9.9)	1	12 (14.8)	1	44 (54.3)	1
**Yes**	5 (11.1)	0.40 (0.16–0.99)	24 (53.3)	0.85 (0.62–1.17)	1 (2.2)	0.22 (0.03–1.74)	5 (11.1)	0.75 (0.28–2.03)	28 (62.2)	1.24 (0.92–1.66)
**Exclusive breastfeeding ^c^**										
**No**	8 (17.0)	1	30 (63.8)	1	4 (8.5)	1	9 (19.2)	1	26 (55.3)	1
**Yes**	15 (19.2)	1.14 (0.53–2.46)	45 (57.7)	0.91 (0.68–1.21)	5 (6.4)	0.75 (0.21–2.66)	8 (10.3)	0.54 (0.22–1.29)	45 (57.7)	1.02 (0.75–1.39)
**Breastfeeding at 1 year**										
**No**	15 (20.0)	1	45 (60.0)	1	7 (9.3)	1	8 (10.7)	1	40 (53.3)	1
**Yes**	8 (16.0)	0.82 (0.38–1.78)	30 (60.0)	1.02 (0.76–1.36)	2 (4.0)	0.42 (0.09–1.96)	9 (18.0)	1.69 (0.70–4.08)	31 (62.0)	1.13 (0.84–1.52)

CI: confidence interval; ne: not estimable; RR: relative risk. ^a^ Maternal smoking during pregnancy or maternal passive smoking exposure during pregnancy. ^b^ Maternal smoking during pregnancy, or maternal passive smoking exposure during pregnancy or child passive smoking exposure during the first year of life. ^c^ Until solid food introduction. ^d^ Maternal smoking during pregnancy, or maternal passive smoking exposure during pregnancy, or child passive smoking exposure during the first 3 years of life. ^e^ Abundant muciparous cells definition: +/++/+++/++++ at birth, and ++/+++/++++ at 1 and 3 years. ^f^ Adjusted for sex. ^g^ Until solid food introduction. ^h^ Odds ratio adjusted for sex, from logistic regression model.

**Table 4 jpm-13-00687-t004:** Association between nasal mucosa cytological composition at birth and development of acute otitis media (AOM), upper respiratory tract infections (URTI), bronchitis, allergy, and bronchial asthma/wheezing during the first year of life (top part of the table) and the first 3 years of life (bottom part of the table).

	AOM		URTI		Bronchitis		Allergy (at 1 Year)/Atopy (at 3 Years)		Bronchial Asthma/Wheezing	
	Cases/tot (%)	RR ^a^	Cases/tot (%)	RR ^a^	Cases/tot (%)	RR ^a^	Cases/tot (%)	RR ^a^	Cases/tot (%)	RR ^a^
1 Year										
**Muciparous** **cells**										
**0**	22/139 (15.8)	1	129/139 (92.8)	1	29/139 (28.1)	1	10/138 (7.3)	1	15/139 (10.8)	1
**+/2+/3+/4+**	10/75 (13.3)	0.82 (0.41–1.62)	67/75 (10.7)	0.96 (0.88–1.05)	19/56 (74.7)	0.90 (0.56–1.44)	4/75 (5.3)	0.74 (0.24–2.27)	8/75 (10.7)	0.99 (0.44–2.22)
**Neutrophils**										
**0/+**	19/110 (17.3)	1	99/110 (90.0)	1	30/110 (27.3)	1	5/110 (4.6)	1	11/110 (10.0)	1
**2+/3+/4+**	13/103 (12.6)	0.70 (0.36–1.33)	97/201 (94.2)	1.04 (0.96–1.13)	28/103 (27.2)	1.02 (0.65–1.59)	9/102 (8.8)	1.96 (0.68–5.68)	12/103 (11.7)	1.08 (0.50–2.33)
**Eosinophils**										
**0**	28/204 (13.7)	1	186/204 (91.2)		56/204 (27.5)	1	13/203 (6.4)	1	21/204 (10.3)	1
**+**	4/10 (40.0)	3.44 (1.55–7.62)	10/10 (100.0)	ne	2/10 (20.0)	0.70 (0.20–2.48)	1/10 (10.0)	1.56 (0.22–10.9)	2/10 (20.0)	2.34 (0.66–8.33)
**Lymphocytes**										
**0**	29/187 (15.5)	1	169/187 (90.4)		52/187 (27.8)	1	13/186 (7.0)	1	21/187 (11.2)	1
**+/2+**	3/25 (12.0)	0.75 (0.25–2.26)	25/25 (100.0)	ne	5/25 (20.0)	0.71 (0.32–1.61)	1/25 (4.0)	0.57 (0.08–4.19)	1/25 (4.0)	0.34 (0.04–2.42)
**Bacteria**										
**0**	25/174 (14.4)	1	162/174 (93.1)	1	49/174 (28.2)	1	13/174 (7.5)	1	19/174 (10.9)	1
**+/2+/3+/4+**	7/39 (18.0)	1.23 (0.57–2.62)	33/39 (84.6)	0.91 (0.79–1.04)	9/39 (23.1)	0.82 (0.44–1.52)	1/38 (2.6)	0.35 (0.05–2.61)	4/39 (10.3)	0.93 (0.34–2.55)
**3 years**										
**Muciparous cells**										
**0**	59/130 (45.4)	1	133/139 (95.7)	1	64/131 (48.9)	1	25/128 (19.5)	1	12/95 (12.6)	1
**+/2+/3+/4+**	28/70 (40.0)	0.88 (0.62–1.24)	68/74 (91.9)	0.96 (0.90–1.03)	29/69 (42.0)	0.86 (0.62–1.19)	14/68 (20.6)	1.02 (0.57–1.82)	5/57 (8.8)	0.69 (0.26–1.85)
**Neutrophils**										
**0/+**	45/100 (45.0)	1	102/110 (92.7)	1	48/101 (47.5)	1	19/99 (19.2)	1	10/75 (13.3)	1
**2+/3+/4+**	41/99 (41.4)	0.91 (0.66–1.25)	98/102 (96.1)	1.03 (0.97–1.09)	45/98 (45.9)	0.98 (0.73–1.32)	20/96 (20.8)	1.05 (0.60–1.84)	6/76 (7.9)	0.56 (0.21–1.49)
**Eosinophils**										
**0**	82/191 (42.9)	1	191/203 (94.1)	1	89/191 (46.6)	1	36/187 (19.3)	1	17/145 (11.7)	
**+**	5/9 (55.6)	1.32 ( 0.72–2.44)	10/10 (100)	ne	4/9 (44.4)	0.93 (0.44–1.97)	3/9 (33.3)	2.03 (0.77–5.32)	0/7 (0)	ne
**Lymphocytes**										
**0**	79/176 (44.9)	1	174/186 (93.4)	1	83/176 (47.2)	1	34/138 (19.8)	1	13/133 (9.8)	1
**+/2+**	7/22 (31.8)	0.69 (0.37–1.31)	25/25 (100)	ne	9/22 (40.9)	0.87 (0.51–1.47)	4/22 (18.2)	0.89 (0.35–2.27)	2/17 (11.8)	1.18 (0.29–4.83)
**Bacteria**										
**0**	74/163 (45.4)	1	166/174 (95.4)	1	79/163 (48.5)	1	31/160 (19.4)	1	15/122 (12.3)	1
**+/2+/3+/4+**	13/36 (36.1)	0.79 (0.50–1.26)	34/38 (89.5)	0.95 (0.85–1.06)	14/36 (38.9)	0.80 (0.52–1.25)	8/35 (22.9)	1.18 (0.60–2.32)	2/29 (6.9)	0.56 (0.14–2.31)

^a^ Adjusted for sex. ne: not estimable.

## Data Availability

The data presented in this study are available on request from the corresponding author.

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
