# Peer review of "Nasal Cytology on 241 Children: From Birth to the First 3 Years of Life and Association with Common Airways Diseases"

_jpm, 2023, doi:10.3390/jpm13040687_

Round 1

Reviewer 1 Report

This is a very interesting work regarding the nasal cytology during the first years of life and its association to several outcomes, such as atopy and respiratory infections.

Nebvertheless, I have some comments to make.

In line 31 please correct the word "realated" into "related".

In line 32 the medical abbreviation "AOM" must be explained.

In lines 73-74 I believe a citation is missing.

Lines 99-111 should be changed into one structured paragraph, and not in "bullet form" as in protocols.

The "Methods" and "Statistical Analysis" sections are presented excellently.

The presentation of the results is sound.

In line 22 please rephrase "is still little investigated".

In line 361 change "if" to "of".

Athough the Discussion is well structured you should add Conclusions to your manuscript.

Reviewer 2 Report

The authors have assessed nasal cytology at birth, first year, and third year of life and correlated various factors with nasal cytology and clinical outcomes at 3 years of age.

The study is of interest, though many factors studied were not of statistical significance. One of the reasons is the dropouts during the study, with COVID playing its part in reducing participation.

General comments:

Spelling and grammar need to be improved

Specific comments:

Please mention what questionnaire was used to assess the clinical outcomes and who developed it. Whether any previously validated questionnaires such as ISAAC/GAN? What language was it administered? It is important to confirm the questions measure what it is supposed to measure and that is the reason a validated questionnaire is critical for a good study.

Caesarean section: 36+17 is not 55. please check the numbers in the table. This cannot be missing data.

Please mention how each of the clinical outcomes was diagnosed in the methodology. For example, how was asthma diagnosed in children 1-3 years of age? Other conditions as well. 

Please define what is ++++, +++ etc

In Table 3 ciliated cells are not discussed

How was the rhiocytogram validated for its reproducibility? Again, this is an extremely important aspect critical for the success of the study. Sputum induction studies are rarely performed as they are very difficult to standardize. Similarly, establishing reproducibility would be critical. Please explain in methods section, how this was done

Line 326 -329: please modify this statement. It needs a microbiome analysis and just a smear is not sensitive enough to comment on bacteria. Please replace microbiological analysis with microbiome analysis

Round 2

Reviewer 2 Report

The authors have addressed the clarifications adequately